# Comprehensive Characterization of the C3HC4 RING Finger Gene Family in Potato (*Solanum tuberosum* L.): Insights into Their Involvement in Anthocyanin Biosynthesis

**DOI:** 10.3390/ijms25042082

**Published:** 2024-02-08

**Authors:** Limin Chen, Yuanming Li, Jinyong Zhu, Zhitao Li, Weilu Wang, Zheying Qi, Dechen Li, Panfeng Yao, Zhenzhen Bi, Chao Sun, Yuhui Liu, Zhen Liu

**Affiliations:** 1College of Agronomy, Gansu Agricultural University, Lanzhou 730070, China; chenlmgsau@foxmail.com (L.C.); lizt1225@163.com (Z.L.); wangwlcn@foxmail.com (W.W.); qzygsau@foxmail.com (Z.Q.); ldc991203@163.com (D.L.); bizz@gsau.edu.cn (Z.B.); sunc@gsau.edu.cn (C.S.); 2Gansu Provincial Key Laboratory of Crop Improvement and Germplasm Enhancement, State Key Laboratory of Aridland Crop Science, Gansu Agricultural University, Lanzhou 730070, China; zhujy_salvare@tom.com (J.Z.); yaopf@gsau.edu.cn (P.Y.); 3College of Horticulture, Gansu Agricultural University, Lanzhou 730070, China; liyuanm@gsau.edu.cn

**Keywords:** StRING-HC gene family, expression profiles, functional analysis, anthocyanin biosynthesis

## Abstract

The C3HC4 RING finger gene (RING-HC) family is a zinc finger protein crucial to plant growth. However, there have been no studies on the RING-HC gene family in potato. In this study, 77 putative *StRING-HC*s were identified in the potato genome and grouped into three clusters based on phylogenetic relationships, the chromosome distribution, gene structure, conserved motif, gene duplication events, and synteny relationships, and *cis*-acting elements were systematically analyzed. By analyzing RNA-seq data of potato cultivars, the candidate *StRING-HC* genes that might participate in tissue development, abiotic stress, especially drought stress, and anthocyanin biosynthesis were further determined. Finally, a *StRING-HC* gene (*Soltu.DM.09G017280* annotated as *StRNF4-like*), which was highly expressed in pigmented potato tubers was focused on. StRNF4-like localized in the nucleus, and Y2H assays showed that it could interact with the anthocyanin-regulating transcription factors (TFs) StbHLH1 of potato tubers, which is localized in the nucleus and membrane. Transient assays showed that StRNF4-like repressed anthocyanin accumulation in the leaves of *Nicotiana tabacum* and *Nicotiana benthamiana* by directly suppressing the activity of the dihydroflavonol reductase (DFR) promoter activated by StAN1 and StbHLH1. The results suggest that StRNF4-like might repress anthocyanin accumulation in potato tubers by interacting with StbHLH1. Our comprehensive analysis of the potato StRING-HCs family contributes valuable knowledge to the understanding of their functions in potato development, abiotic stress, hormone signaling, and anthocyanin biosynthesis.

## 1. Introduction

The RING (Really Interesting New Gene) finger protein was first discovered as TFIIA in the oocytes of the *Xenopus laevis* [1], which was one of the TFs of RNA polymerase II and widely present in a variety of plants, especially in young and active plant tissues [2], and was also the host of many *Xanthomonas*. Inhibiting the expression of TFIIA-related genes can improve plant resistance to phytopathogens [3,4]. The RING finger structure is cysteine-rich, has eight metal ligand sites, and contains four pairs of cysteine/histidine residues that can bind two zinc ions to form a tetrahedral structure [5]. The general formula of the domain of RING finger is Cys-X2-Cys-X(9-39)-Cys-X(1-3)-His-X(2-3)-Cys/His-X2-Cys-X(4-48)-Cys-X2-Cys [6]. The difference between the two typical RING fingers C3H2C3 (RING-H2) and C3HC4 (RING-HC) lies in the position of the metal ligand 5, which can either be a cysteine or a histidine [6]. The RING-HC proteins are further subdivided into RING-HCa and RING-HCb at amino acid numbers between ml7 and ml8 [7,8]. In addition, other alteration types, such as RING-C2, RING-D, RING-V, RING-G, and RING-S/T were also discovered [9]. The RING finger domain prefers the interaction between two proteins in terms of function [10].

Most RING-HCs are E3 ligases that serve as carriers for the transport of ubiquitin molecules to target proteins and regulate the hydrolysis process of proteins in plants [11]. Being E3 ubiquitin ligases, the RING-HC proteins are also involved in plant growth and development, stress response, and disease resistance [12]. Research on the involvement of RING-HCs in plant growth has primarily focused on photoperiod, root, flower and fruit development [13]. During the photoperiod, the RING-HC protein COP1 in *Arabidopsis thaliana* can inhibit the expression of *HY5*, which is a bZIP transcription factor that directly induces photosynthesis under dark conditions, thereby inhibiting the establishment of light morphology under light conditions [14]. The RING-HC proteins are related to organ development, for instance, XBAT32 regulates the growth of lateral roots [15], and RGLG3/4 regulate the jasmonic acid (JA)-inhibited root elongation in *Arabidopsis thaliana* [16]. During flower and fruit development, the RING1A, SINAL7, and ORTH family members in *Arabidopsis thaliana,* and HAF1 and FRRP1 in rice (*Oryza sativa*) affect flower development by regulating flowering time [17,18,19,20,21]. During fruit development, NbZFP1 regulates fruit growth in tobacco (*Nicotiana benthamiana*) [22], and RAD5A is involved in bud and horn fruit growth, as well as seed maturation in *Arabidopsis thaliana* [23].

Adversity stress can limit the growth of plants. The RING-HC proteins have also been implicated in response to biotic and abiotic stresses. For example, AtVBP1 and AtAIRP3/LOG2 positively participate in the abscisic acid (ABA)-dependent defense mechanism under drought stress in *Arabidopsis thaliana* [24,25], while AIR1 and DTR1 in peppers (*Capsicum annuum*) negatively regulate drought response in an ABA-dependent manner [26,27]. EBS1 in *Arabidopsis thaliana* is involved in bicarbonate tolerance [28], and RZFP1 in cabbage is involved in salt, dehydration, and cold stress [29]. Under biotic stress, ATL9 in *Arabidopsis thaliana* and RHC1 in rice promote immune response by ubiquitinating and hydrolyzing inhibitory proteins against pathogens [30,31].

With the continuous in-depth study of the RING-HC proteins, the role of RING-HCs in the regulation of secondary metabolites is gradually revealed, especially in flavonoid and anthocyanin accumulation [32,33]. Anthocyanin is a flavonoid compound that confers plants a rich color due to its light absorption [34]. These bright colors serve the purpose of attracting birds and insects, which aid in the pollination and dispersal of pollen and seeds for reproduction [35]. And anthocyanin can also enhance stress resistance such as low temperature, drought, and UV resistance, as well as the efficient scavenging of free radicals [34]. The synthesis pathway of anthocyanin is the shikimate pathway [36]. Anthocyanin biosynthesis is affected by two types of genes [37]. One includes the structural genes contained in the plant itself that regulate anthocyanin synthesis; these directly encode enzymes involved in anthocyanin formation, such as *CHS*, *CHI*, *F3H*, *DFR*, *F3′H*, *F3′5′H*, *ANS*, *UFGT*, etc. [37]. The other includes the regulatory genes that are not directly involved in the biosynthesis of anthocyanins, but affect the biosynthesis of anthocyanins by regulating the expression of structural genes [38]. Many gene families of TFs contribute to the regulation of anthocyanin biosynthesis, with the MYB, bHLH, and WD40 TFs being the most significant and extensively investigated ones [39]. A previous study found that these three transcription factors form a MYB-bHLH-WD40 (MBW) complex to regulate anthocyanin synthesis [40]. Liu et al. found that StAN1 and StbHLH1 co-regulated the anthocyanin biosynthesis [41]. And StMYB3 negatively regulated anthocyanin biosynthesis, which clusters with AtMYB3 and AtMYB4 of four subfamilies of the R2R3-MYB gene family in *Arabidopsis thaliana* [42,43]. StMYBATV has previously been shown to be a negative regulator of anthocyanins in potato [44,45]. Both of these interacted with StbHLH1 to inhibit the synthesis of anthocyanin [45].

Within the RING-HC gene family, the most well-known COP1 not only mediated the ubiquitin-dependent degradation of many processes such as the synthesis of alkaloids and phenylpropanoids [46,47], but was also associated with anthocyanin biosynthesis in many species [48,49,50,51,52,53]. For example, LcCOP1 interacts with LcHY5 and LcMYB1 to negatively regulate anthocyanin accumulation in litchi (*Litchi chinensis*) [53], and the synthesis of anthocyanin in the injection site of the overexpression of *PbCOP1* in pear is also reduced [49]. In apple, MdMIEL1 ubiquitinated and degraded MdMYB1 via the 26s proteasome pathway, and led to decreased anthocyanin content in apple calli [54].

Potato is widely planted in the world because of its drought tolerance, high adaptability, and stable yield. The accumulation of anthocyanin in potato tubers leads to the red, purple, blue, and other colors of potato skin and flesh [55]. Meanwhile, a large number of accumulated anthocyanins makes the antioxidant capacity of pigmented potato 3–4 times higher than ordinary potato [56]; the starch content is also low, and the ability to prevent mutation, neurodegenerative, cardiovascular, and cerebrovascular diseases, and osteoporosis, and inhibit tumor cells is also enhanced [56,57]. Due to their enhanced nutritional value and economic benefits, pigmented potatoes have garnered considerable attention [58]. RING-H2 proteins have been studied for improving plant growth, resistance, and tuber quality in potato. For example, the overexpression of *StRFP2* increased physiological and biochemical indexes such as plant height, and enhanced drought tolerance [59]. It found that *SbRFP1* was conducive to the synthesis of starch in potato plant leaves and changed the shape of starch granules; meanwhile, SbRFP1 acted as E3 ubiquitin ligase to reduce the accumulation of starch and sugar in potato tubers by the ubiquitination and degradation of the β-amylase gene *StBAM1* [60]. However, studies on RING-HC proteins in potatoes have rarely been reported. In the present work, the whole genome level of potato was analyzed to identify 77 putative members of the RING-HC gene family. Subsequently, the location on the chromosome, the structure, and duplication events of *RING-HC* members, and the phylogenetic development and conserved domains of the proteins were systematically investigated. Moreover, the *StRING-HC* gene expression in different tissues and under abiotic stress of double monoploid (DM) potato were analyzed using RNA-seq data of the PGSC database. Furthermore, the RNA-seq data were applied to investigate the expression profiles of *StRING-HC* genes in three pigmented potato clones and in drought-tolerant and drought-sensitive cultivars under drought stress. Finally, a candidate gene that might be involved in anthocyanin biosynthesis in potato tubers was verified. This study provides a theoretical basis for further exploring the functions of the StRING-HC gene family and identifying RING-HC gene family members in other plant species.

## 2. Results

### 2.1. Identification of StRING-HC Proteins

The RING finger conserved domain sequence downloaded from the Pfam database was used as a BLAST query against the PGSC database by using HMM 3.1. The SMART and NCBI (CDD) were used to screen the sequence with the RING finger domain. Finally, a total of 77 putative members of the StRING-HC gene family, including 22 *StRING-HC1*, 27 *StRING-HC2*, and 28 *StRING-HC3*, were obtained. The 77 predicted StRING-HC proteins differ in their physicochemical properties and amino acid sequences. The number of amino acid residues of the StRING-HC proteins ranged from 330 (Soltu12G028180) to 14,298 (Soltu02G001760), with the molecular weights between 12.41 kDa (Soltu12G028180) and 534.77 kDa (Soltu02G001760), and the pI values varied in the range of 4.27 (Soltu06G002260) to 9.57 (Soltu02G010670). Predictions of the subcellular localization of the StRING-HC protein were used with CELLO v.2.5, and the results showed that 60 family members were localized in the nucleus, nine in the plasma membrane, and eight in the extracellular (Appendix A).

### 2.2. Chromosomal Distribution, Phylogenetic Analysis, and Classification of StRING-HCs

The 77 identified *StRING-HC* genes were unevenly distributed in 11 chromosomes. Chromosome 2 contained the largest number of genes with 12 *StRING-HC* genes, while chromosomes 8 and 11 had the lowest number of genes with 3 each. Moreover, some *RING-HC* members were also distributed in two ends of chromosomes, such as chromosomes 7 and 9 (Figure 1). In addition, we downloaded the chloroplast and mitochondrial genome information of potato and found that there were no members of the *StRING-HC* gene family.

The multiple alignment of 77 StRING-HC and 74 AtRING-HC and the construction of an unroot phylogenetic tree were performed to investigate the evolution of RING-HC gene family in potato and *Arabidopsis thaliana* using MEGA7.0 software. The results showed that these 151 RING-HCs were divided into three subgroups (RING-HC1, RING-HC2, and RING-HC3). Twenty-two StRING-HCs and fifteen AtRING-HCs belonged to the RING-HC1 subgroup. Twenty-seven StRING-HCs and twenty-nine AtRING-HCs were assigned to the RING-HC2 subgroup. Twenty-eight StRING-HCs and thirty AtRING-HCs were included in the RING-HC3 subgroup (Figure 2).

### 2.3. Gene Structure and Conserved Motifs of StRING-HC Members

MEGA 7.0 software was used to construct an unrooted phylogenetic tree based on the amino acid sequences of the 77 identified StRING-HCs, as shown in Figure 3A. The introns and exons of *StRING-HC* family members were analyzed to understand the structure of these *StRING-HC* genes (Figure 3B). The analysis showed that the number of introns and exons of these *StRING-HC* genes varied greatly. Of the 77 *StRING-HC* genes, 18 members only contained introns, 11 members contained 1–3 introns, 29 members contained introns between 4 and 10, and 19 members contained more than 10 introns. Moreover, performing MEME was analyzed on the conserved protein domain of the StRING-HC family. Twenty diversified motifs were identified and denominated Motif 1–Motif 20 (Figure 3C and Appendix A). Some motifs were only present in the specific subfamilies, for example, Motif 6 only appeared in the StRING-HC1 subfamily, four motifs (Motifs 3, 7, 9, and 12) only existed in the StRING-HC2 subfamily, and six motifs (Motifs 8, 10, 11, 13, 16, and 18) were only distributed in the StRING-HC3 subfamily. Motifs 19 and 20 were present in the StRING-HC1 and StRING-HC2 subgroups. Motifs 5 and 14 were present in the StRING-HC2 and StRING-HC3 subgroups, whereas Motif 1s and 2 were widely distributed in all three StRING-HC subgroups, which included 16 and 15 amino acids, respectively. The RING-HC domain was composed of Motif 4, Motif 1, and Motif 2. Motif 1 was mainly present at the N terminal of the RING-HC domain, and Motif 2 primarily appeared at the C terminal of the RING-HC domain. In general, the motifs in a subgroup were relatively conserved in composition and the genetic structures were similar. The results further demonstrated the reliability of the evolutionary analysis.

### 2.4. Gene Duplication and Collinearity Analysis

Gene duplication is crucial in the expansion of family members and functions. Collinearity analysis identified that two genes situated in chromosome 2 were a pair of tandem duplicated genes (2/77, 2.60%) (Figure 1A). Twelve pairs (18/77, 23.38%) of segmental duplication events occurred on chromosomes 1, 2, 3, 6, 7, 9, 10, and 12 (Figure 4).

The ratio between non-synonymous mutations (Ka) and synonymous mutations (Ks) was used as an indicator to measure whether selection pressure acts on genes. A Ka/Ks value of 1 represents neutral evolution, Ka/Ks < 1 illustrates that the gene undergoes purification selection, and Ka/Ks > 1 shows positive selection. The Ka/Ks value of the tandem duplicated genes was 0.3311, while the Ka/Ks ratios of segmental duplicated genes were between 0.1465 and 0.4811, with an average value of 0.3239 (Appendix A). The Ka/Ks values of the duplications were all less than 1, illustrating that the evolution of these genes was influenced by purification selection.

We further analyzed orthologous genes of the *RING-HC* gene family to explain the potential evolution of these genes in tomato, Arabidopsis, cabbage, rice, and maize (Figure 5A). As a result, 53, 27, 28, 7, and 6 pairs of orthologous genes were identified, respectively, and the corresponding ranges of the Ka/Ks values were 0.0188–0.8028, 0.0624–0.3015, 0.0686–0.2196, 0.1013–0.1464, and 0.1184–0.2973 (Figure 5B and Appendix A). Each ratio was smaller than 1, indicating these genes evolved under purified selection.

### 2.5. Promoter CIS-Acting Element Analysis

Promoters can influence the growth of plants and resist adversity by regulating gene expression. The Plant CARE was used to analyze the 2000 bp upstream sequences of the transcription start site (TSS) of each gene to study the *cis*-acting elements of the promoter region of *StRING-HC*s. The results showed that apart from the common CAAT-box and TATA-box, 18 other *cis*-acting elements concerned with light response, hormone response, growth and development, and abiotic stress were identified. In particular, 42 genes contained the G-box element involved in light response, 65 genes contained the abscisic acid responsive element ABRE, 21 genes contained the CAT-box element relevant to plant growth and development, and 24 genes contained the MBS element linked to drought stress response. MYB and MYC elements can be combined by MYB and MYC transcription factors to have relevance with many biological processes in plants, and in our work, 77 genes contained MYB- and MYC-responsive elements (Figure 6 and Appendix A).

### 2.6. Expression Profiles of StRING-HC Genes in Different Tissues

Using RNA-seq data downloaded from the PGSC database, this study analyzed the tissue-specific expression patterns of the *StRING-HC* gene in multiple tissues of DM potato, including roots, shoots, leaves, stolons, petioles, flowers, petals, stamens, carpels, sepals, tubers, immature fruits, and mature fruits (Figure 7 and Appendix A). More than 32% of *StRING-HC* genes (25/77) had high expression levels in all tissues with FPKM > 5, while 14.29% (11/77) of them showed lower expression with FPKM < 2. Notably, *Soltu02G034590*, *Soltu05G010500*, *Soltu06G023120*, *Soltu09G020810*, *Soltu09G022530*, and *Soltu12G028210* were highly expressed in all 13 tissues (FPKM > 15), suggesting their important involvement in the organ development of potato. Certain *StRING-HC* genes had a tissue-specific expression; for example, *Soltu10G004220* in the *StRING-HC1* subfamily was highly expressed in shoots, and *Soltu11G021190* in subfamily 2 and *Soltu09G017280* in subfamily 3 were only highly expressed in stamens and mature flowers.

### 2.7. Expression Profiles of StRING-HC Genes under Abiotic Stresses and Hormone Treatments

The expression profiles of *StRING-HC* genes in DM potato under salt, heat, and drought stress were analyzed using transcriptome data downloaded from PGSC. The results showed that 17, 26, and 23 genes were differentially expressed (FPKM > 1, |log_2_FC| > 1) under salt, mannitol, and heat treatments, respectively. Of these, 15, 25, and 13 *StRING-HC* genes were upregulated, respectively. Among them, *Soltu07G016550*, *Soltu10G004220*, *Soltu10G026870*, and *Soltu11G001010* were differentially expressed under the three stress treatments. Twenty genes showed differential expression under two stresses and ten genes were only differentially expressed under a single abiotic stress (Figure 8 and Appendix A).

Moreover, the expression patterns of *StRING-HC* genes under hormone treatments (BAP, ABA, IAA, and GA3) were analyzed using RNA-seq data downloaded from PGSC. The results showed that 15, 3, 4, and 9 *StRING-HC* genes responded to BAP, IAA, GA3, and ABA treatments (FPKM > 1, |log_2_FC| > 1), respectively (Figure 8 and Appendix A). Of these, 12, 2, 4, and 6 genes were upregulated, respectively. Moreover, *Soltu10G004220* was upregulated under the treatment of three plant hormones (IAA, GA3, BAP) with FPKM > 20, and the fold change was 2.61, 2.23, and 2.81, respectively.

In order to further obtain the candidate *StRING-HC* genes that responded to drought stress, we conducted RNA-seq on the drought-sensitive cultivar ‘Atlantic’ (A) and the drought-tolerant cultivar ‘Qingshu No. 9’ (Q) under drought treatment. The results showed that 14 genes displayed low expression or undetectable expression, and 8 genes showed differential expression profiles under drought stress (Figure 8 and Appendix A). Among them, *Soltu05G010500* was upregulated at the early flowering stage and full-blooming stage, *Soltu03G002940* and *Soltu09G017690* were upregulated at the flower-falling stage, while *Soltu02G017860*, *Soltu05G018370*, and *Soltu05G002540* were downregulated at the flower-falling stage (FPKM > 1, |log_2_FC| > 1). *Soltu07G017450* was downregulated at all three growth stages, with the fold changes being −2.45, −2.64, and −2.43, while *Soltu12G029640* was upregulated at all three growth stages, with the fold changes being 2.03, 2.18, and 2.86, respectively. By combining the RNA-seq analyses on DM potato under mannitol stress, we further discovered that *Soltu09G017690* and *Soltu12G029640* were not only highly upregulated in DM under mannitol stress, but also showed significantly high expression in Q (FPKM > 1, |log_2_FC| > 1), indicating the possible involvement of these genes in the response to drought stress, which is worthy of further investigation.

### 2.8. Expression Profiles of StRING-HC Genes in Pigmented Potato Clones

To analyze the expression pattern of the *StRING-HC* genes in pigmented potato tubers, the tuber flesh of three potato clones (CK: yellow-fleshed potato; H: red-fleshed potato; and Z: purple-fleshed potato) at three developmental stages (S1: tuberization stage; S2: tuber bulking stage; and S3: tuber maturation stage) was chosen for RNA-seq sequencing. Our results revealed that 11 *StRING-HC* genes showed low or undetectable expression (FPKM < 1), while 8 genes showed higher expression in pigmented tubers (FPKM > 5) (Figure 9 and Appendix A). At the three stages, 26, 15, and 9 genes were differentially expressed compared to CK, of which 19, 6, and 2 genes were upregulated in pigmented flesh, respectively. During the tuberization stage, 11 genes were upregulated in H and Z, and 3 genes (*Soltu01G027650*, *Soltu07G017530*, *Soltu07G019810*) were downregulated. *Soltu02G008910* was upregulated in H and Z, with the fold changes being 7.75 and 8.31, respectively, while *Soltu01G027650* was downregulated in H and Z, with the fold changes being −2.36 and −4.36, respectively. During the tuber bulking stage, *Soltu09G017280* was upregulated in H and Z with FPKM > 30. *Soltu02G017730* and *Soltu10G004220* were downregulated in H and Z, with the fold changes being −1.93 and −2.31, and −1.58 and −4.51, respectively. During tuber maturation stage, *Soltu09G017280* was upregulated in H and Z with FPKM > 40, while *Soltu07G017360* and *Soltu10G004220* were downregulated in H and Z, with the fold changes being −2.61 and −2.91, and −1.72 and −3.28, respectively. Notably, the expression of *Soltu07G019810* was less than 10 in H at the S1 and S2 stages with the flesh being white/yellow, while the FPKM value reached up to 40.91 at the S3 stage when the flesh turned to red, which was 4.32 and 6.16 times higher than those at the S1 and S2 stages, respectively. Apart from that, another candidate gene, *Soltu09G017280*, showed a significantly higher expression in the potato flesh of H and Z during all three growth periods compared with that in Y, suggesting the possibility of their involvement in the anthocyanin biosynthesis of pigmented potato tubers, which is worthy of further study.

We further analyzed the correlation relationship between Soltu09G017280 and anthocyanin pathway genes, such as CHS-1 (Soltu09G028560), CHS-2 (Soltu05G023610), CHI-1 (Soltu05G022280), CHI-2 (Soltu05G001950), F3′5′H (Soltu11G020990), DFR (Soltu02G024900), ANS (Soltu08G026700), F3H (Soltu02G023850), and UFGT (Soltu09G017160), as well as the anthocyanin-regulating transcription factors AN1 (Soltu10G020850), bHLH1 (Soltu09G019660), MYB3 (Soltu05G004700), and MYBATV (Soltu12G023200) [41,45]. The results showed that Soltu09G017280 had no strong correlation with MYB3 and MYBATV (R^2^ = 0.45), but had a positive correlation with DFR, AN1, bHLH1, with an R^2^ of 0.84, 0.72, and 0.74, respectively, indicating that Soltu09G017280 might be involved in the anthocyanidin biosynthesis of potato tubers (Figure 10).

### 2.9. Cloning and Functional Analysis of the StRING-HC Gene

The CDS fragment of *Soltu09G017280* (678 bp) was cloned by using the cDNA of RS3 as a template. The homology between the obtained gene and the reference gene sequence was 97.2%. The protein sequence of Soltu09G017280 had a high homology with the RNF4-like protein in pepper and tobacco (Appendix A), and thus Soltu09G017280 was annotated as StRNF4-like.

To explore the involvement of StRNF4-like in the anthocyanin biosynthesis of potato tubers, the yeast two-hybrid assay was conducted to probe the interaction of StRNF4-like with the anthocyanin-regulating transcription factors StAN1, StMYB3, StMYBATV, and StbHLH1 [41,45]. *StRNF4-like* was cloned into the pNC-GBKT7 vector, and these transcription factors were all cloned into the pNC-GADT7 expression vector. The results showed that only the combination of StRNF4-like-BD and StbHLH1-AD could grow on both defective media, and no interaction was detected between StRNF4-like and the other three MYBs (Figure 11A). The result indicated that StRNF4-like interacts with StbHLH1, suggesting that its involvement in anthocyanin biosynthesis in potato tubers might be by ubiquiting StbHLH1, which requires further experimental verification.

Then, *StRNF4-like* and *StbHLH1* were ligated into the subcellular localization expression vector pNC-121-SubN to detect their subcellular localization, respectively, and pNC-121-SubN was used as control. These expression vectors were injected into the leaves of *Nicotiana benthamiana*, and a GFP fluorescence signal was observed under a laser confocal microscope after 2 days. The results showed that the green fluorescence signals were observed in both the cytoplasm and nucleus of leaves being injected with pNC-121-SubN and pNC-121-SubN-StbHLH1, while specific fluorescence signals were present in the nucleus of leaves when injected with pNC-121-SubN-StRNF4-like (Figure 11B), indicating that StRNF4-like was localized in the nucleus, while StbHLH1 was localized in the cell membrane and nucleus. It was hypothesized that StRNF4-like may interact with StbHLH1 at the nucleus, which needs further investigation.

Previous studies have found that StAN1 was a major transcription factor that regulated anthocyanin biosynthesis in potato tubers. It could interact with StbHLH1 to promote anthocyanin accumulation [41]. In order to further explore the function of StRNF4-like in anthocyanin biosynthesis, the color assay was performed in *Nicotiana tabacum*. The results showed that the tobacco leaf injected with StAN1 and StbHLH1 showed significant anthocyanin accumulation, with an intense red pigmentation observed in tobacco leaves, while light-red color was observed in the leaves when StAN1 and StbHLH1 were co-infiltrated with StRNF4-like (Figure 12A), suggesting that StRNF4-like was able to suppress anthocyanin accumulation. Subsequently, a dual luciferase assay was carried out to further clarify that StRNF4-like negatively regulates anthocyanin biosynthesis. The LUC/REN ratio was calculated 3 days after the injection of different combinations. The ratio of LUC/REN in the control group (StAN1 + StbHLH1 + EV and *StDFR* promoters) was 4.55, while the ratio of LUC/REN in the experimental group (StAN1 + StbHLH1 + StRNF4-like and *StDFR* promoters) was 2.57. There was a significant difference between the two groups (*p* < 0.01), indicating that StRNF4-like inhibited the activation of the *StDFR* promoter activated by the transcriptional activators StAN1 and StbHLH1 (Figure 12B). The results further confirmed that StRNF4-like was a negative regulator of anthocyanin biosynthesis, which can inhibit the expression of the key structure gene *StDFR*, thereby inhibiting the synthesis of anthocyanins.

### 2.10. Validation of qPCR for StRNF4-like

Through the qPCR verification of *StRNF4-like* in three potato clones, we found that the relative expression level of *StRNF4-like* was consistent with the changed trend of RNA-seq, and the correlation coefficient R^2^ value was 0.956 (Appendix A). In our previous study, eight genes and transcription factors related to anthocyanin synthesis were selected for qPCR analysis at three tuber developmental stages of three potato clones. It was found that the *R*^2^ value between the qPCR results and the RNA-seq data was 0.8838, and the two methods were highly correlated [45]. This shows that the results of this study were consistent with the previous studies.

## 3. Discussion

The C3HC4 RING finger proteins (RING-HCs) in plants are pivotal for the processes of growth and development, and resist adverse environmental factors [61,62,63]. In this study, 77 *StRING-HC*s were identified, and a phylogenetic analysis of the RING-HCs genes in potato and Arabidopsis were carried out to speculate their evolution and possible function. We also conducted comprehensive analyses on the chromosome distribution, gene structure, conserved motif composition, and gene duplication events of *StRING-HCs.* Moreover, the expression profiles of *StRING-HC* family members in different tissues, as well as under abiotic and exogenous hormone treatments, were investigated of DM potato. More importantly, the candidate genes which responded to drought stress and might be associated with the anthocyanin biosynthesis of potato tuber were screened based on RNA-seq in two cultivars with differential drought tolerance and in three clones with differently colored tubers. Finally, a candidate gene that might be concerned with anthocyanin biosynthesis was further investigated.

### 3.1. Phylogenetic Analysis and Evolution of StRING-HCs

Phylogenetic comparative analysis determines the function of direct relative and collateral homologous genes from the functions of known genes based on the fact that members of the same subfamily have a common origin and conserved function [12]. Previous studies found that 91 RING-HCs in poplar were divided into seven different subfamilies [64]. Similarly, 204 RING-HCs in flax were divided into three main subgroups [8]. In this study, we identified 77 StRING-HCs, which were primarily categorized into three subgroups based on phylogenetic analysis. The classification of StRING-HCs was consistent with gene structure characteristics and conserved motifs. These members were unevenly distributed on the 11 chromosomes. The RING-HC structure was Cys-X2-Cys-X(9-39)-Cys-X(1-3)-His-X(2-3)-Cys/His-X2-Cys-X(4-48)-Cys-X2-Cys [6,10]. The analysis of the conserved protein domains revealed that Motif 4, Motif 1, and Motif 2 constituted the RING-HC structure, and that the seven cysteines (cys) and one histidine (his) contained therein, combined with the two zinc ions, intersected to form a tetrahedral structure and thus carry out its function [6]. A phylogenetic comparative analysis has found that *DRIP1/2* negatively regulated drought stress by targeting DREB2A in Arabidopsis, a transcription factor that controls the expression of water deficit-induced genes to the hydrolysis process of 26S proteasomes [65]. In our work, *Soltu06G034450* was directly homologous to *DRIP1/2* in subfamily 1, indicating that the gene may have similar functions as *DRIP1/2*. *RGLG2/3/4* were clustered in the same branch with four genes (*Soltu09G020590*, *Soltu05G024060*, *Soltu03G025320*, and *Soltu02G033570*) in subfamily 2. *RGLG2* mediates ubiquitination to negatively regulate drought stress signaling in Arabidopsis [66], and *RGLG3/4* positively respond to various biological processes in Arabidopsis, such as the inhibition of root growth, pathogen localization, and wound response by regulating JA signaling [16]. It is likely that these genes were also involved in these processes. *AtAIRP2* positively regulates ABA-mediated drought stress by participating in ABA-mediated root inhibition growth and stomatal closure [67]. *Soltu10G022750* was homologous with *AtAIRP2*, so it may also be related to these processes.

### 3.2. Expansions of the StRING-HC Gene Family in Potato

The tandem duplication and segmental duplication mainly drive the generation of new family members and new functions in plant evolution [68]. In this study, there were 13 pairs of duplication genes identified in potato, with 2.6% being tandem, and the number of segmental duplication genes accounted for 23.38%. This implied that the evolution of the *StRING-HC* gene was dominated by fragment replication, and some duplicated genes are likely to be generated through gene duplication.

Duplication events will lead to the diversification of functions between a pair of genes and to the loss or acquisition of certain functions by some members [69]. In our study, the expression of some duplicate genes was consistent, while others were opposite; for example, the segmental duplication gene pairs *Soltu07G019810* and *Soltu10G004220* were upregulated under salt and mannitol stresses, whereas *Soltu02G017340* was downregulated, and *Soltu10G004220* was upregulated under heat stress. The tandem duplicated pairs *Soltu02G010660* and *Soltu02G010670* also showed different expression patterns in tissues, where the *Soltu02G010660* was expressed in 13 tissues, while *Soltu02G010670* was only specifically expressed in petals, indicating that *Soltu02G010660* may obtain some new functions during duplication events and evolution.

### 3.3. The StRING-HCs Respond to Abiotic Stress

It has been reported that the *RING-HC* genes have also been shown to resist abiotic stress [12]. There were 21 differential genes expressed in DM under two or more abiotic stresses, while 6 genes were expressed in opposite patterns under different stresses, and 10 genes were expressed differently only under single stress, indicating that the responses of *StRING-HC* to abiotic stresses were various. By further analyzing the expression profiles of these members in the A and Q cultivars under drought stress combined with the expression under mannitol treatment, it was found that two genes (*Soltu09G017690* and *Soltu12G029640*) were upregulated in the drought-resistant cultivar Q and under mannitol treatment, while the expression profiles of three genes (*Soltu10G004220*, *Soltu05G010500*, and *Soltu05G002540*) were opposite in Q and under mannitol treatment; these genes were likely to be involved in the drought stress response. For example, *Soltu12G029640*, which was in the same cluster as *AtRMA1*, increased significantly under Q and mannitol treatments, with a fold change greater than 2, while *AtRMA1* was reported to play a role with UBC32, a ubiquitin-conjugating enzyme that regulated plant response to multiple drought stresses and improved plant drought resistance [70], suggesting that *Soltu12G029640* may also play a role in responding to drought stress

Studies have found that *RING-HC* gene family members not only participate in drought stress, but also have a regulatory effect on abiotic stresses such as high temperature and salt [10]. For example, *OsSADR1* was homologous to *At1G47570*, which negatively regulated salt and drought stress by regulating the protein level of the target protein. In this study, *Soltu06G009790* in the same cluster as *At1G47570* was downregulated under salt and mannitol treatment, suggesting that *Soltu06G009790* may be involved in abiotic stress [71]. The above results revealed that these *StRING-HC* genes respond to abiotic stresses, and further study of these genes would be useful.

### 3.4. Identification of the StRING-HCs Associated with Anthocyanin Biosynthesis

It has been reported that COP1, which is an important member of the RING-HC gene family, has an important influence on plant photomorphogenesis [14]. As an E3 ubiquitin ligase, COP1 could also ubiquitinate and degrade MYB transcription factors at the post-translation level to affect the anthocyanins’ biosynthesis in apple, crabapple, and other species [49,50,52,72]. Apart from COP1, several other RING-HC members also involved in anthocyanin synthesis through the ubiquitination pathway. For example, MdMIEL1 negatively regulated anthocyanin accumulation through the ubiquitinated degradation of the MdMYB1 protein [54]. MdMIEL1 also interacted with MdMYB308L to promote the ubiquitination degradation of MdMYB308L and negatively regulated cold resistance and anthocyanin accumulation [54]. The RING-type E3 ligase CtBB1 in *Carthamus tinctorius* can ubiquitinate CtbHLH41, which reduces the expression of *DFR* promoted by CtbHLH41, thereby negatively regulating the accumulation of anthocyanin and HYSA (hydroxysafflor yellow A) [73].

In our work, the candidate *RING-HC* members that might be associated with anthocyanin biosynthesis, and the interaction between the candidate *RING-HC* genes and anthocyanin-regulating factors in potato tubers, were also probed. The results showed that eight genes were highly expressed in pigmented potato flesh; among them, *Soltu07G019810* was differentially expressed in the flesh of the red clone during the critical stage of color accumulation. *Soltu09G017280* (*StRNF4-like*) was highly expressed in pigmented flesh at all three stages, showing a high correlation with several anthocyanin pathway genes and the transcription factor StAN1 and StbHLH1. Thus, the function of StRNF4-like in anthocyanin biosynthesis was further investigated. Our previous work found that potato anthocyanin activator StAN1 and co-factor StbHLH1, which are highly expressed in colored tubers, functioned together to upregulate the anthocyanin biosynthetic genes, leading to anthocyanin accumulation in tobacco leaves, while repressors StMYB3 and StMYBATV, which are also highly expressed in colored tubers, could competitively interact with the co-factor StbHLH1 and repress anthocyanin biosynthesis, acting in the negative feedback regulation mechanism on anthocyanin biosynthesis during pigmented potato tuber development [41,45]. StRNF4-like has a high correlation with StAN1 and StbHLH1 according to expression levels, and it functioned as a repressor to negatively regulate anthocyanin biosynthesis via a tobacco transient assay in our work, both of which suggest a possible regulatory network between StRNF4-like and these anthocyanin synthesis-related TFs. *StRNF4-like* showed an overlapping expression profile with StbHLH1 in the pigmented potato tuber, Y2H, and transient assays of tobacco indicated that StRNF4-like interacted with StbHLH1 to negatively regulate anthocyanin biosynthesis. The above results indicate a feedback regulatory mechanism on the fine-tuning of potato tuber pigmentation during tuber development. RNF4-like has been reported as a DNA damage-responsive protein that played a role in DNA repair functions [74]. This is the first time that the E3 ligase RNF4 was discovered to be involved in anthocyanin biosynthesis in potato tubers, and to possibly confer a repressive function by ubiquitinating StbHLH1. The mechanism is worthy of further investigation.

## 4. Materials and Methods

### 4.1. Identification of StRING-HCs in Potato

The identification of RING-HC members in potato and Arabidopsis was performed by downloading the whole protein sequences of potato and Arabidopsis from PGSC version 6.01 (http://solanaceae.plantbiology.msu.edu/pgsc_download.shtml, accessed on 12 June 2023) and TAIR (http://plants.ensembl.org/index.html, accessed on 12 June 2023), respectively. The Hidden Markov Model (HMM) profiles of the RING-HC (PF00097, PF13923, PF13920, and PF15227) domain were downloaded from the Pfam database (http://pfam.xfam.org, accessed on 25 June 2023) [75], and HMMER 3.1 software (http://hmmer.org/download.html, accessed on 25 June 2023) was used to search and screen protein sequences that contain the RING-HC domain. Finally, the candidate members with RING-HC domains were manually verified through SMART (http://smart.embl-heidelberg.de, accessed on 28 June 2023) and the NCBI Conservative Domain Database (CDD).

### 4.2. Analysis of the Physicochemical Properties of StRING-HCs

A physicochemical analysis of StRING-HCs proteins was conducted on ExPasy (https://web.expasy.org/protparam/, accessed on 5 July 2023) [76], including amino acid numbers, molecule numbers (MW), and theoretical isoelectric point (pI). CELLO v.2.5 (https://mybiosoftware.com/cello-v-2-5-subcellular-localization-predictor.html, accessed on 10 July 2023) [77] was used to predict the subcellular localization of StRING-HCs.

### 4.3. Phylogenetic Analysis and Classification of RING-HCs

The alignment of the sequences of 77 StRING-HCs and 74 AtRING-HCs was carried out using ClustalW version 2.0.10 software based on the maximum likelihood method; the mode was Poisson model, and 1000 bootstrap iterations were performed. Phylogenetic trees were constructed using the MEGA7.0 software [78].

### 4.4. Sequence and Structural Characterization of StRING-HCs

The conserved protein motif of StRING-HC was analyzed using MEME version 5.0.4 (http://alternate.meme-suite.org/tools/meme, accessed on 25 July 2023) to analyze the properties of conserved domains. The maximum number of motifs was set to 20, the ideal motif width was set to 6–50 amino acid residues, and the rest of the parameters were used as default [79]. The structures of *StRING-HC*s members were plotted using GSDS 2.0 (http://gsds.cbi.pku.edu.cn/, accessed on 28 July 2023) [80].

### 4.5. Duplication of Genes and Chromosomal Localization

Based on the description of the chromosome position in PGSC, the chromosome location of *StRING-HC* gene was plotted using MapChart version 2.32 [81]. Meanwhile, the StRING-HC genes were also mapped to the potato chloroplast and mitochondrial genomes [82,83,84]. A collinearity analysis and visualization were conducted using MCScanX v1.5.1 [85] and Circos v0.69 [86] to investigate StRING-HCs duplication events. Moreover, the non-synonymous substitution rate (Ka) and synonymous substitution rate (Ks) for each pair of duplicated genes were further estimated by using KaKs Calculator 2.0 [87].

### 4.6. Analysis of Promoter Sequences of StRING-HCs

The promoter sequence of the transcriptional start site upstream (−2000 bp) of the *StRING-HC*s was obtained based on the PGSC datebase, and then the *cis*-acting elements of the *StRING-HC*s promoters were analyzed using PlantCARE (http://bioinformatics.psb.ugent.be/webtools/plantcare/html/, accessed on 6 August 2023).

### 4.7. Plant Materials and Treatments

Three different pigmented potato clones, namely, Y (yellow-fleshed and white-skinned potato), R (red-fleshed and red-skinned potato), and P (purple-fleshed and purple-skinned potato), and two different drought-resistant cultivars, ‘Atlantic’ (a drought-sensitive cultivar) and ‘Qingshu No. 9’ (a drought-tolerant cultivar), were used as experimental materials [45,88]. The above experiments collected samples from three individual plants and set three biological replicates. All experimental materials were planted in the experimental base of the Dingxi Academy of Agricultural Sciences in Gansu Province.

It should be noted that ‘Atlantic’ (A) and ‘Qingshu No. 9’ (Q) were planted under rain shelters. All treatments before seedling emergence received equal irrigation. After 15 days of seedling emergence, the plants under drought stress were no longer supplied with water during the subsequent growth stage [88]. Finally, the flesh of three pigmented potato clones were taken at three developmental stages (S1: tuberization stage; S2: tuber bulking stage; and S3: tuber maturation stage), and the leaves of the two varieties were collected at three growth stages (T1: early flowering stage; T2: full-blooming stage; T3: flower-falling stage), and finally stored in a refrigerator at −80 °C after liquid nitrogen quick freezing.

The subcellular localization and dual luciferase experiments were performed using *Nicotiana benthamiana* grown to 5–7 mature leaves as experimental materials, and leaf coloration experiments selected *Nicotiana tabacum* with suitable growth (6-leaf stage) as experimental materials.

### 4.8. Extraction of RNA, Reverse Transcription of cDNA, and Quantitative Real-Time PCR

The total RNA was extracted from the flesh of three different pigmented potatoes and the leaves of two different drought-tolerant cultivars of potatoes collected under drought stress using the RNA extraction kit (Cat. No. 9769, TaKaRa, Shiga, Japan). RNA integrity and concentration were determined via agarose gel electrophoresis and spectrophotometer (Nanodrop Technologies, Wilmington, DE, USA) measurements, respectively. The synthesized cDNA first strand was performed with the FastKing RT kit (Cat. No. KR116, Tiangen, Beijing, China) with gDNase to eliminate genomic DNA pollution. Quantitative real-time PCR (qPCR) was performed on CFX96 (Bio-Rad, Hercules, CA, USA) with the Tiangen’s SuperReal PreMix Plus (Cat. No. FP205, Tiangen, Beijing, China) kit, the experimental process was carried out according to the above kit instructions, and three biological replicates were set. Reaction was performed with *StEF-1α* as an internal reference gene [89], and the Ct (2^−ΔΔCt^) method was used to calculate the gene relative expression. The figure was drawn with the use of Origin 2021. The primers were designed and synthesized using Bioengineering (Shanghai, China), and the detailed primer list is in Appendix A.

### 4.9. Expression Pattern Analysis of StRING-HCs in Potato

The expression of the *StRING-HC* gene family in different tissues (including mature fruit, immature fruit, roots, leaves, shoots, tubers, stolons, petioles, sepals, flowers, stamens, carpels, and petals) of DM potato was analyzed with Illumina RNA-seq data downloaded from PGSC, as well as under abiotic stress (drought treatment: 260 µM mannitol, 24 h; salt treatment: 150 mM NaCl, 24 h; heat treatment: 35 °C, 24 h) and hormone treatment (abscisic acid treatment: 50 µM ABA, 24 h; gibberellin treatment: 50 µM GA3, 24 h; indole acetic acid treatment: 10 µM IAA, 24 h; 6-benzylamino purine treatment: 10 µM BAP, 24 h). The heat map was drawn with TBtools software v2.019. Furthermore, high-throughput RNA sequencing (Biomarker Technologies Corporation, Beijing, China) was performed on tubers of three different pigmented potato clones at three developmental stages, and on leaves of two cultivars at three growth stages under drought stress, to investigate the expression profiles of the *StRING-HC* gene family; after sequencing was completed, the results were uploaded on NCBI (Project ID PRJNA541096 and PRJNA782081) [88].

### 4.10. Cloning of the StRING-HC Gene

The CDS sequence of the candidate gene *Soltu09G017280* was amplified using cDNA as a template, and the primers were: F: 5′-ATGAGCACTCAAGACAAAAGGGC-3′, R: 5′-TCAACTTGTGGCAGGAAGAT-3′. PCR amplification was performed in a 25 μL amplification system with high-fidelity polymerase (Cat. No. R010A, TaKaRa, Japan). The amplified products were cloned into pNC-AEn-TOPO, pGBKT7, and pNC-121-SubN vector.

### 4.11. Subcellular Localization

The amplified *StRNF4-like* and *StbHLH1* were reconstituted through NC cloning [90] into the subcellular expression vector pNC-121-SubN and transferred into the competent cells DH5α. Then, the recombinant plasmids were transferred to competent cells GV3101. Agrobacterium solution was injected into the leaves of *Nicotiana benthamiana*, and after 2 days, a laser confocal microscope (LSCM 800, Carl Zeiss, Jena, Germany) was used to observe whether in the injection area there was fluorescence at the excitation light of 488 nm.

### 4.12. Yeast Two-Hybrid Assays

The amplified *StRNF4-like* was reconstituted into the yeast expression vector pGBKT7. The anthocyanin-regulating transcription factors StAN1, StMYBATV, StMYB3, and StbHLH1 were cloned into the pNC-GADT7 expression vector [41,45]. The recombinant plasmids were co-transformed into yeast-competent cells AH109 and coated on the corresponding yeast medium. A single colony was picked up after inversion culture at 30 °C for 3–4 days, and the obtained bacterial solution was enlarged, diluted, and dripped in the SD/-Leu/-Trp and the SD/-Ade/-His/-Leu/-Trp medium, respectively. The yeast growth was observed on the medium after 4–5 days.

### 4.13. Tobacco Transient Expression Experiment

*Soltu09G017280* (*StRNF4-like*) was ligated into pNC-cam2304-MCS35S vector and transferred into GV3101 competent cells, and the transient assays were carried out in *Nicotiana tabacum* and *Nicotiana benthamiana*. The coloration of leaves was observed after 3–5 days when injecting StAN1, StRNF4-like, StbHLH1, StRNF4-like + StbHLH1, StRNF4-like + StAN1 + StbHLH1, and StAN1 + StbHLH1 + EV in *Nicotiana tabacum*, respectively. In the dual luciferase assay, the promoter of StDFR prom-3-StDFR in pGreenII 0800-LUC vector, and StAN1 and StbHLH1 in pSAK277 and pHEX2 were used [41]. The StAN1 + StbHLH1 + EV and *StDFR* promoter was co-infiltrated into the leaves of *Nicotiana benthamiana* as the control, while StAN1 + StbHLH1 + StRNF4-like and *StDFR* promoter were injected as the experimental group. Three days after injection, the LUC and REN activities were measured using the kit (Cat. No. E1910, Promega, Madison, WI, USA), and then the LUC/REN ratio was calculated.

### 4.14. Data Statistical Analysis

The qPCR and Luc experimental data were analyzed using the statistical software package SPSS 22.0. Data presented as the mean (±SE) of three biological replicates were compared using one-way ANOVA, followed by the calculation of the least significant difference (LSD) in *p* < 0.05.

## 5. Conclusions

The 77 putative *StRING-HC* genes were identified in this study, their physicochemical properties and positions in cells were predicted, and these genes were grouped into three subunits based on phylogenetic characteristics. The distribution of these members on the 11 chromosomes was uneven. It was found that the expansion of the StRING-HC gene family was mainly due to segmental duplication events via collinearity analysis. Synteny analysis revealed that potato has 27, 53, 28, 7, and 6 pairs of genes that were orthologous with Arabidopsis, tomato, cabbage, rice, and maize, respectively, which have all evolved under strong purifying selection. Many light-, hormone- and stress-responsive *cis*-acting elements were discovered in the promoters of *StRING-HC*s. Furthermore, the expression patterns of *StRING-HC*s in different tissues and abiotic stresses of DM potato, as well as in pigmented potato clones and drought-tolerant tetraploid potato cultivars, were analyzed to find genes related to spatial distribution, stress, and anthocyanin biosynthesis. More importantly, a StRING-HC protein (StRNF4-like) that negatively regulates anthocyanin biosynthesis in potato tubers was functionally confirmed.

## Figures and Tables

**Figure 1 ijms-25-02082-f001:**
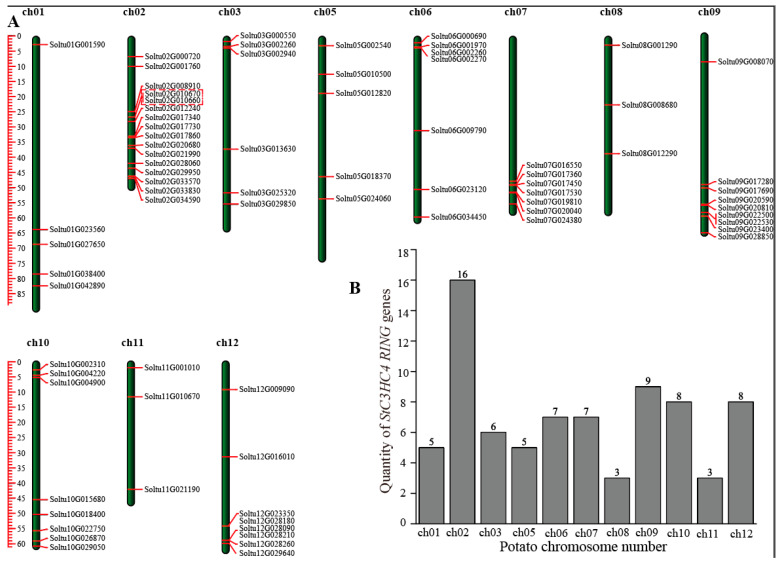
The 77 *StRING-HC* genes were distributed in 11 chromosomes. (**A**) The distribution of the *StRING-HC* gene on 11 chromosomes. The red boxes mark tandem duplicated genes. (**B**) The distribution numbers of *StRING-HC* on each chromosome of the potato. Soltu in the figure represents Soltu.DM.

**Figure 2 ijms-25-02082-f002:**
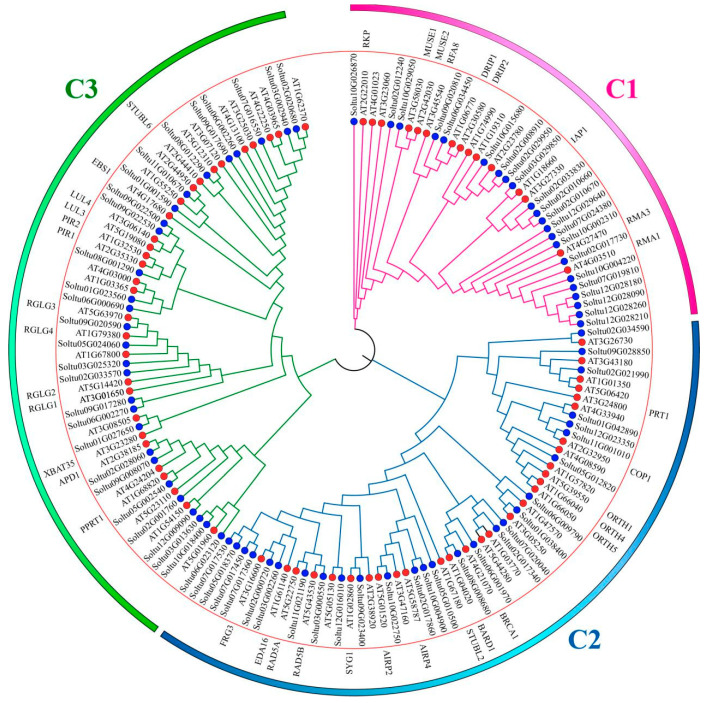
Phylogenetic tree of *Arabidopsis thaliana* and potato RING-HCs. The three different subgroups are denoted with three colored lines (red, green, and blue). The red circles represent AtRING-HCs and the blue circles refer to StRING-HCs.

**Figure 3 ijms-25-02082-f003:**
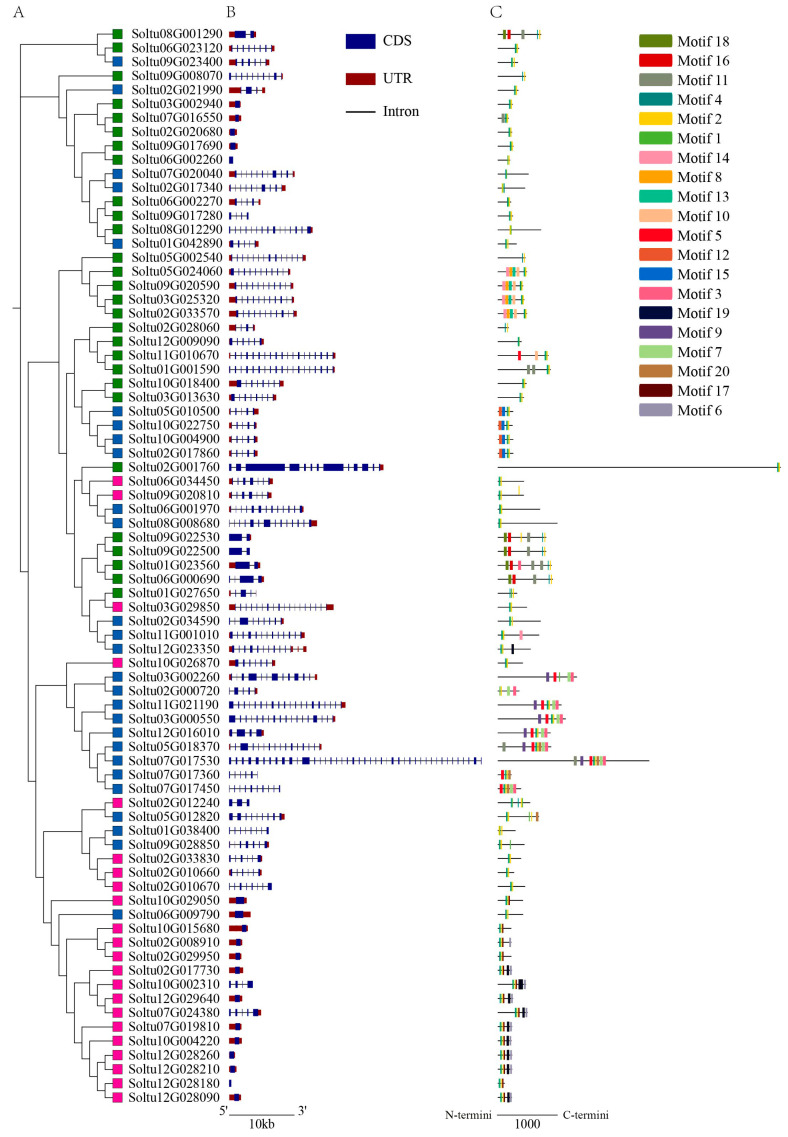
The evolutionary relationship, gene structure, and conserved motif analysis of the StRING-HC gene family. (**A**) Evolutionary tree of StRING-HCs. Subfamilies 1, 2, and 3 are denoted by the pink box, blue box, and green box, respectively. (**B**) Exon/intron structure of the *StRING-HC*s gene. Blue boxes indicate exons and black lines indicate introns. Red boxes indicate 5′UTR and 3′UTR. (**C**) Conserved motif distribution of StRING-HC. The 20 differently colored boxes represent the 20 specific motifs.

**Figure 4 ijms-25-02082-f004:**
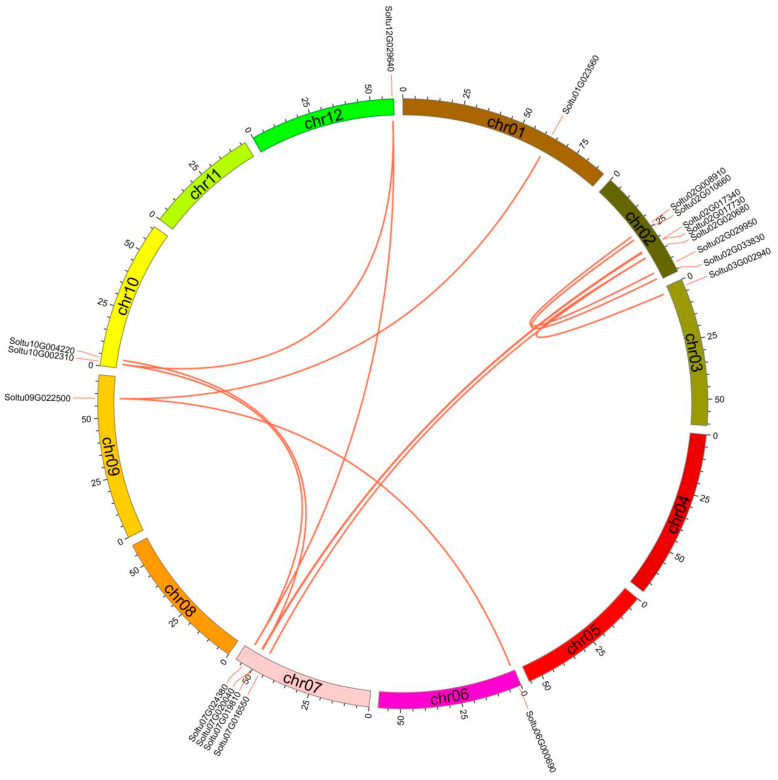
Duplication events of the *StRING-HC* gene family. The red line indicates the homology of *StRING-HC* genes, and chromosome numbers are displayed at the bottom of each chromosome.

**Figure 5 ijms-25-02082-f005:**
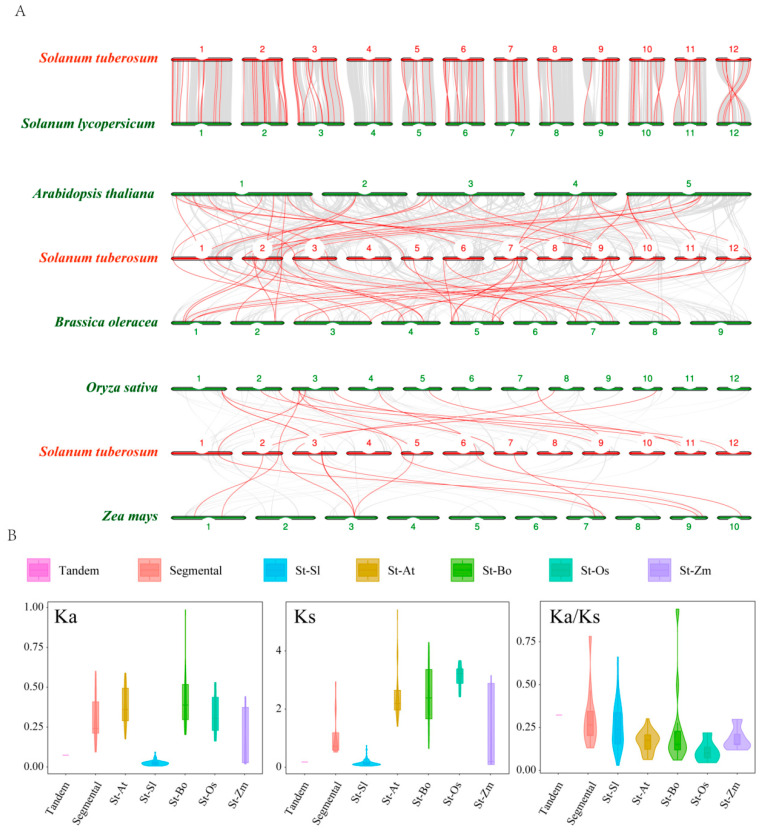
The synteny relationships of the *RING-HC* genes between tomato, Arabidopsis, cabbage, rice, maize, and potato. (**A**) The red line indicates homology between the *StRING-HC* genes and the *AtRING-HC* genes, *SlRING-HC* genes, *OsRING-HC* genes, *BoRING-HC* genes, and *ZmRING-HC* genes. (**B**) The average values of Ka, Ks, and Ka/Ks of the duplicated genes are shown. The horizontal axes indicate tandem duplication (Tandem), segmental duplication (Segmental), and the duplication between potato and Arabidopsis (St-At), tomato (St-Sl), cabbage (St-Bo), rice (St-Os), and maize (St-Zm), respectively.

**Figure 6 ijms-25-02082-f006:**
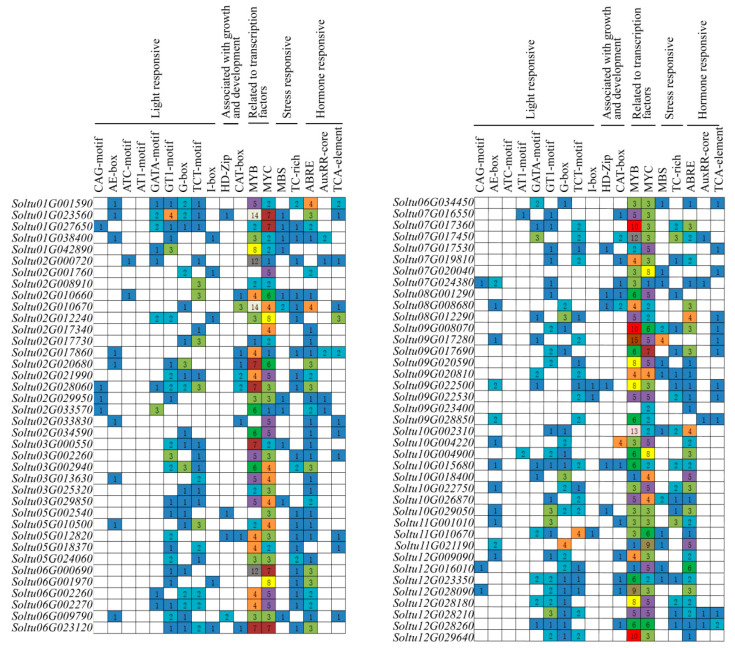
Putative *cis*-acting regulatory elements in the *StRING-HC* gene promoters. The number of *cis*-acting elements is represented by the number in the different colored block.

**Figure 7 ijms-25-02082-f007:**
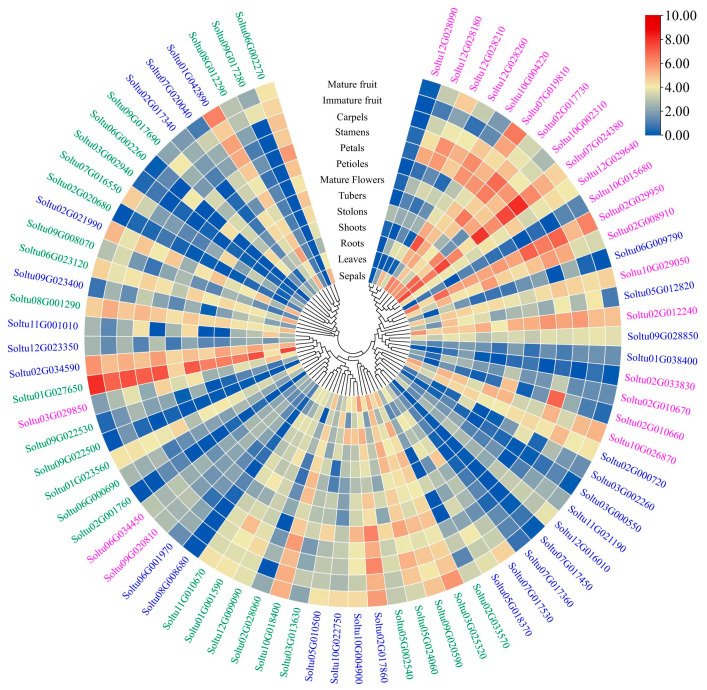
Expression of *StRING-HC*s in different tissues. The colored blocks from blue to red represent the expression levels of genes normalized with log_2_FPKM and log_2_FC, respectively. The gene members in subfamilies 1, 2, and 3 are marked with pink, blue, and green color, respectively.

**Figure 8 ijms-25-02082-f008:**
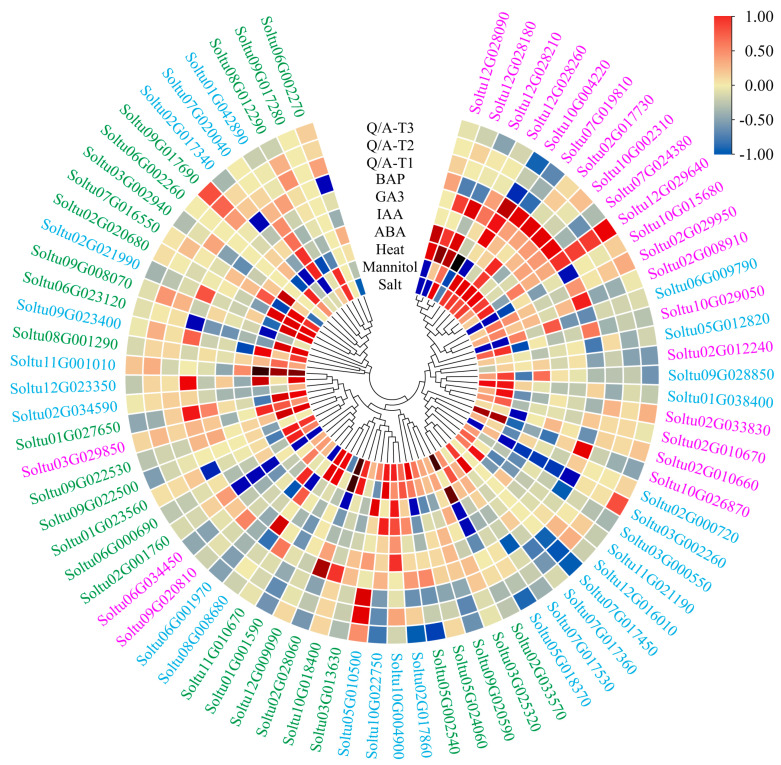
The expression profiles of *StRING-HC* genes in four hormones (IAA, ABA, GA3, and BAP), under multiple stress in double monoploid (DM) potato, and in the drought-sensitive cultivar ‘Atlantic’ (A) and drought-tolerant cultivar ‘Qingshu No. 9’ (Q) under drought stress at three growth stages (T1: early flowering stage; T2: full-blooming stage; and T3: flower-falling stage). The colored blocks from blue to red represent the expression levels of genes normalized with log_2_FPKM and log_2_FC, respectively. The gene members in subfamilies 1, 2, and 3 are marked with pink, blue, and green color, respectively.

**Figure 9 ijms-25-02082-f009:**
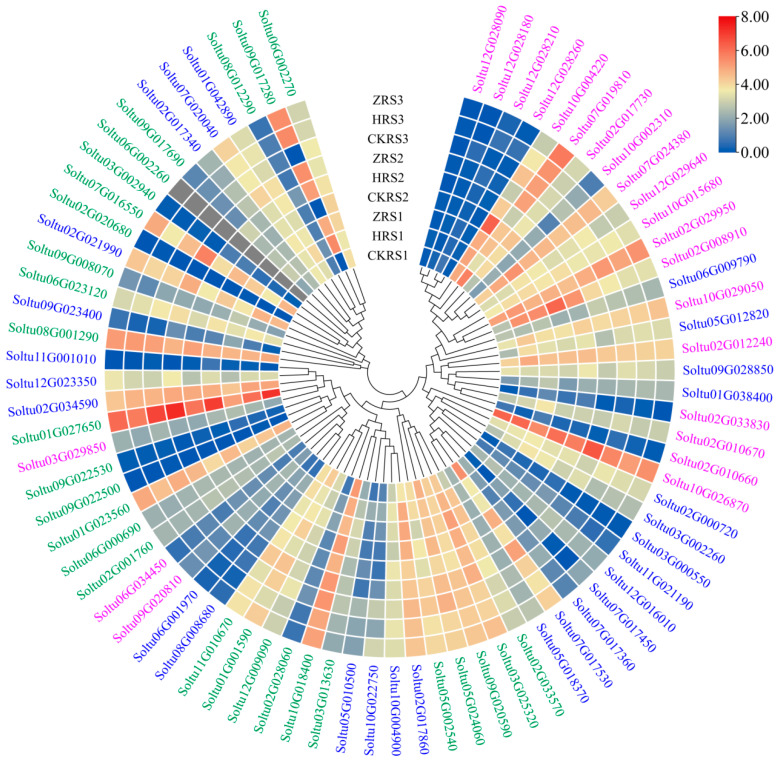
The expression profiles of *StRING-HC* genes in the white and pigmented potato tubers. CK, H, and Z represent the yellow clone (Y), red clone (R), and purple clone (Z). S1, S2, and S3 represent three developmental stages. S1 is the tuberization stage, S2 is the tuber bulking stage, and S3 is the tuber maturation stage. The gene members in subfamilies 1, 2, and 3 are marked with pink, blue, and green color, respectively.

**Figure 10 ijms-25-02082-f010:**
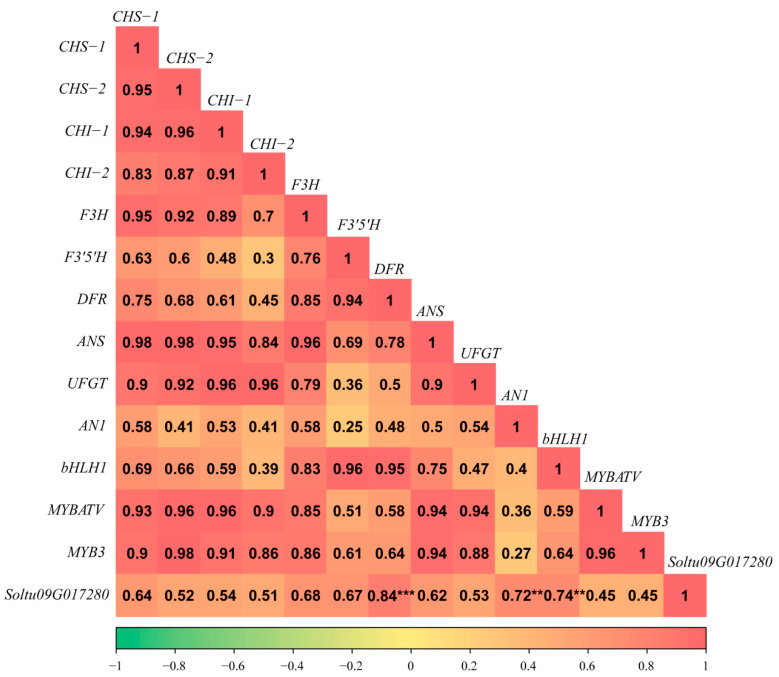
Correlation analysis of *Soltu09G017280* with genes associated with anthocyanin synthesis. The values in each box represent the correlation coefficient between the expression levels of the corresponding row and column genes in the three tuber development stages of the three colored potato clones. The * symbol represents the level of significance (*p* < 0.05). The greater the number of the * symbols, the higher the significant difference.

**Figure 11 ijms-25-02082-f011:**
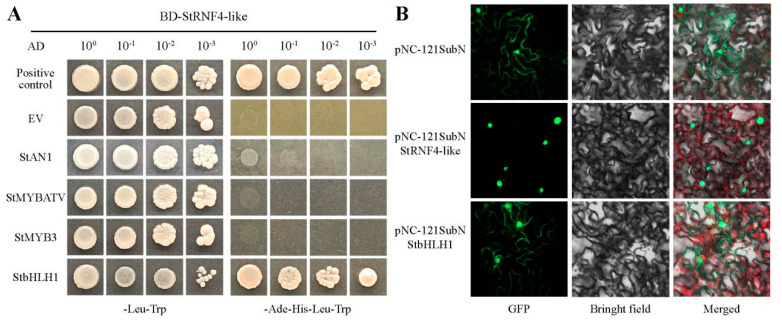
The functional analysis of StRNF4-like. (**A**) The interaction of StRNF4-like and StbHLH1 on yeast. Yeast cells, co-transformed with BD-StRNF4-like and AD-StAN1, AD-StMYBATV, AD-StbHLH1, and AD-StMYB3, were grown on -2SD (-Leu, -Trp) and -4SD (-Leu, -Trp, -His, -Ade) medium, respectively. The negative control was BD-StRNF4-like with AD-EV, and the positive control was BD-StbHLH1 with AD-StAN1. (**B**) The subcellular localization of StRNF4-like and StbHLH1 (scale bars: 20 μm). Red light was autofluorescence of chloroplasts.

**Figure 12 ijms-25-02082-f012:**
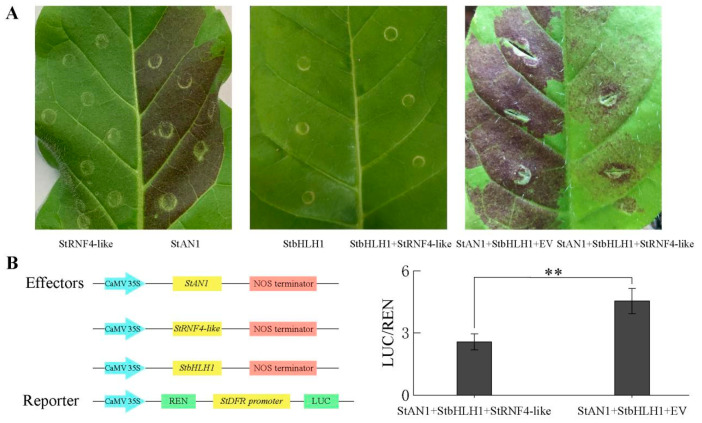
Function of StRNF4-like in anthocyanin synthesis. (**A**) Tobacco transient expression experiment. StbHLH1, StRNF4-like, StAN1, StRNF4-like, StAN1 + StbHLH1 + StRNF4-like, and StAN1 + StbHLH1 + EV were injected on both sides of tobacco, respectively. (**B**) The dual luciferase assay. *StDFR*Pro-LUC + StAN1 + StbHLH1 + EV was the control group, and *StDFR*Pro-LUC + StAN1 + StbHLH1 + StRNF4-like was the experimental group. The data are the mean of three independent biological replicates (±SE). ** represent extremely significant difference (*p* < 0.01).

## Data Availability

Data are contained within the article and Appendix A.

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
