# Peer review of "Comprehensive Characterization of the C3HC4 RING Finger Gene Family in Potato (Solanum tuberosum L.): Insights into Their Involvement in Anthocyanin Biosynthesis"

_ijms, 2024, doi:10.3390/ijms25042082_

Round 1

Reviewer 1 Report

Comments and Suggestions for Authors

ijms-2831951Comprehensive characterization of the C3HC4 RING finger gene family in potato (Solanum tuberosum L.): insights into their involvement in anthocyanin biosynthesis

This is an interesting manuscript that provides new information on the regulation of anthocyanin biosynthesis in potato (Solanum tuberosum L.) by C3HC4 RING finger gene family. This manuscript employs several different methods to characterize this gene family, including bioinformatics, transcriptomic and several other experimental approaches (Y2H, tobacco transient expression).

To the best of my knowledge, molecular regulation of anthocyanin biosynthesis has not been sufficiently studied before in potato. There is also a lack of knowledge on C3HC4 RING finger gene family in plants. Therefore, this manuscript is relevant for IJMS and the data are novel. However, several specific important edits are required to improve the manuscript for further processing. 

 Major edits:

- It is necessary to carefully explain in the manuscript why StAN1, StMYB3, StMYBATV and StbHLH1 have been chosen for interaction identification in Y2H experiments. It is known that multiple transcription factors control anthocyanin biosynthesis in plants both for negative and positive regulation. Therefore, provide and discuss the available literature information on what is known in particular on these 4 factors, namely StAN1, StMYB3, StMYBATV and StbHLH1 or their homologs in Arabidopsis. It should be done  in the Introduction and/or Discussion.

- Explain why you did not generate a cDNA library and screen it for interactions in the Y2H assays? You have chosen only four specific TFs for possible interactions, while there are multiple TFs in anthocyanin biosynthesis. StRNF4-like could interact with some other proteins related to anthocyanin biosynthesis or other processes.

- Please mention a reasoning here what your data suggest that StAN1, StMYB3, StMYBATV and StbHLH1 could  be functionally related/interact with StRNF4?

- It is  required to provide literature information what is currently known on the regulation of anthocyanin biosynthesis in plants in general and by transcription factors, and in potato in particular. For example, MBW complex, what are its components, positive and negative regulation, what TF are involved. It should be done in the Introduction and/or Discussion.

- It is necessary to include in the Introduction a brief description of anthocyanins: anthocyanins – functions of anthocyanins in plants; the practical benefits of anthocyanins in medicine etc; anthocyanin biosynthesis pathway – some general information.

- I think that it is necessary to analyze StRNF4-like gene expression by qRT-PCR in order to support and verify the RNAseq data.

- Figure 7 is not readable. It should be divided into two Figures, and the Figure size should be increased to make them clear for understanding.

Minor:

- Check all abbreviations, e.g. in Abstract TF was not mentioned in full (line 23).

- Check where gene and gene family names should be in italics. For example, line 16 and line 108-109,

StRING-HC gene family, including 22 StRING-HC1, 27 StRING-HC2, and 28 StRING-HC3 All should be in italics.

Comments on the Quality of English Language

English is generally good, but moderate editing of the English language is required. Please carefully proofread the manuscript. I recommend that a native speaker or a professional editing service would proofread the manuscript. Please note that these are only several examples and I recommend that more work on English should be done.

Several examples for correction:

- line 22, Correct “It’s localized in the nuclear, Y2H assays showed that…” TO “It is localized in the nucleus, and Y2H assays showed that….”

- line 84. Correct “barren tolerance”. What type of tolerance did you mean? It is not clear.

- line 277, Correct “Of  which,  Soltu02G008910…..”

- line 348. Correct “It could interact with to promote anthocyanin accumulation [46].” Interact with what?

- line 439 Correct “drought coercion”. Coercion is not appropriate word here.

Author Response

Reviewer 1

Reviewer 2 Report

Comments and Suggestions for Authors

Manuscript ijms-2831951 presents interesting and important data on the C3HC4 RING finger gene family in potato. The authors provided detailed information about the genes of this family and their regulation under various stresses. The data presented by the authors will be of interest to a wide range of researchers. However, the manuscript contains many errors and polemical details that require correction.

Major remarks:

1. The results from the first part of the “Results” section (sections 2.1. – 2.4.) are not discussed in any way, which raises the question: How important are the results presented in these sections? What do these results add to knowledge about the C3HC4 RING finger gene family? And would removing these results improve the manuscript? Provide a more detailed discussion of Figures 1, 2, 3, and 4 in the Discussion section, comparing the data presented with the literature.

2. Designations are given for genes that cannot be studied without additional explanations from the authors. Indicate in detail which database the numbers used are for, so that readers can easily navigate to any nucleotide or amino acid sequence used in the manuscript.

Minor remarks:

1) Line 15: replace “subunits” with “subgroups” or “clusters”.

2) Lines 21 and 101: “Soltu.DM.09G017280” is used only here. In other cases, this gene is designated “Soltu09G017280”.

3) Line 22: “It’s localized in the nuclear” - In the context, “It” means “StRING-HC gene”. The protein that is the product of this gene is localized in the nucleus.

4) Line 23: TF – in the abstract section, provide the full name of the abbreviation.

5) Lines 36-38: Change the sentence. The present version of this proposal does not indicate the transition from animals to plants. What does “the regulation of growth and plant pathogen” mean?

6) Lines 105-108: Have the genomes of the mitochondria and chloroplast of potato been studied? Add this information to Figure 1.

7) Line 110: “amino acid lengths” replace with “amino acid sequences”. “the number of amino acids” change to “the number of amino acid residues”.

8) Table S1 – “MW (Kda)” replace with “MW (Da)”.

9) Page 6: Questions and comments about Figure 3

1) Is the phylogenetic tree based on nucleotide or amino acid sequences? Indicate the parameters of the phylogenetic analysis and provide a discussion of where the gene/protein tree falls on the same branches (Figure 3a), which are very different in Figures B and C.

2) Figure 3C: “5’ and 3’” scale – amino acid sequences do not have 5’ and 3’ ends. They have N- and C-termini.

3) Motifs: explain the characteristics of these motives. Do these motifs have functional significance? What does the presence of certain motives indicate? Are there any of these motifs that are responsible for assigning proteins to zinc fingers?

10) Table S5 duplicates Figure 6.

11) Lines 532-534: Provide different symbols for the six different stages of potato development.

Author Response

Reviewer 2

Reviewer 3 Report

Comments and Suggestions for Authors

Review of the manuscript entitled " Comprehensive characterization of the C3HC4 RING finger gene family in potato (Solanum tuberosum L.): insights into their involvement in anthocyanin biosynthesis ". I find the work interesting and valuable. I have a few thoughts that require consideration before accepting the work for publication.

1.      Complete the abstract of the paper with information about the methods used in the study.

2.      The keywords overlap with the information in the title of the article. This should be corrected.

3.      At the end of the introduction section, instead of describing the scope of the research, state the purpose for which the work was conducted. It is advisable to include a graphical summary with the article, showing the sequence of the research work in relation to the assumed purpose of the research. A description of the plant material used for the research appears only in Section 4.7 (Plant material and treatments) of Chapter . Materials and methods.

4.       Fig. 9: What do the values in the graph mean? *- Determine these significance levels.

5.      L522-535. How many replicates were used in the study? Did the authors take the conduct of the study into account in the study series?

6.      L541. Nanodrop Technologies, USA. Indicate the model, manufacturer and headquarters of the manufacturer.

7.      l538-544. In how many replicates per sample were the analyses conducted?

8.      L553. Give model, manufacturer, and headquarters of manufacturer (high throughput RNA sequencing).

9.      L569. ..a laser confocal microscope... Give model, manufacturer, and headquarters of manufacturer.

10.   L579. Give the name of the medium

Author Response

Reviewer 3

Round 2

Reviewer 1 Report

Comments and Suggestions for Authors

A minor edit that can be done at the proofs stage:

- please add a comment to the manuscript based on your response to my questions 2 and 3 in my previous review.

Comments on the Quality of English Language

- English requires improvements in newly added text (yellow) in the revised manuscript.

Author Response

reviewer 1

Reviewer 2 Report

Comments and Suggestions for Authors

The authors have significantly improved the manuscript. I think that the revised manuscript can be accepted for publication in the IJMS after a minor revision.

 Minor remarks:

1. To distinguish experimental data from predicted data, authors should label results obtained through bioinformatics analysis as “predicted” and “putative”. For example, in line 15, “77 StRING-HCs were identified in the potato genome” would be more correctly “77 putative StRING-HCs were identified in the potato genome” since there is no experimental data demonstrating that products of these genes function as C3HC4 RING finger proteins.

2. The first sentence of the Introduction section (lines 34–36) remains very difficult to understand. I suggest that the authors split it into 2-3 sentences with clarifications that would more clearly demonstrate the information that the authors want to show in this part of the manuscript. In addition, in the references [2-4] listed here, I did not find any mention of the association of TFIIA with plant growth. And I did not find the authors’ answer to my question: “What does “the regulation of growth and plant pathogen” mean?" As I understand it, “plant pathogen” means bacteria of the genus Xanthomonas. In this case, it is better to use “plant-pathogen interaction” or “phytopathogens”.

3. “we searched the literature related to potato mitochondrial and chloroplast genomes, and also downloaded the related information, and did not find the C3HC4 RING finger members on mitochondria and chloroplasts in this study” - Add the information about the absence of C3HC4 RING finger members genes in the mitochondrial and chloroplast genomes of potato in section “2.2. Chromosomal Distribution, Phylogenetic Analysis, and Classification of StRING-HCs".

Author Response

reviewer 2
